# Identification of lubricant viscosity to minimize the frictional impact of colonoscopy on colonic mucosa
Naoto Watanabe[1,2], Ryohei Hirose [1,2] ✉, Hiroshi Ikegaya[3], Katsuma Yamauchi [1,2], Hajime Miyazaki[1,2], Takuma Yoshida[1], Risa Bandou[3], Ken Inoue[1], Osamu Dohi[1], Naohisa Yoshida[1], Takaaki Nakaya[2] & Yoshito Itoh[1]

Applying a lubricant to the colonic mucosa and reducing the dynamic friction coefficient (DFC) between the endoscopic shaft and colonic mucosa may reduce colonoscopy invasiveness. However, the ideal lubricant viscosity remains unknown. Here, we developed a DFC measurement model integrating samples of colonic mucosa from forensic autopsy specimens into a simulated bowel bend and determined the low-friction lubricant viscosity that minimizes the DFC. Carboxymethyl cellulose, xanthan gum, hydroxyethyl cellulose, sodium alginate, and sodium polyacrylate aqueous solutions of various concentrations were used as lubricants. Low-friction lubricants minimized the load on the colonic mucosa during colonoscope insertion and reduced the total endoscopy insertion time. The highest correlation was between the DFC and the lubricant viscosity at a shear rate of 100 1/s. The lowest DFC was almost constant at approximately 0.09, irrespective of the chemical composition of the lubricant, and the low-friction lubricant viscosity (100 1/s) was 0.031–0.086 (median: 0.059). The viscosities of conventional colonoscopic lubricants were suitable for lubricating the anorectal skin owing to their low DFC, but too high for lubricating the colonic mucosa because of their high DFC. The utilization of the low-friction lubricants with the optimal viscosity can reduce the stress on colonic mucosa during colonoscopy.

Colonoscopy is a crucial examination for detecting colorectal diseases, such as neoplastic and inflammatory diseases[1,2]. In upper gastrointestinal endoscopy, the narrowing of the diameter of the endoscope has made it possible to perform less invasive examinations with less burden, even for high-risk patients such as the elderly[3,4]. However, colonoscopy is more invasive than upper gastrointestinal endoscopy, with a higher incidence of complications, such as pain and perforation during endoscopy insertion[5–11]. Therefore, colonoscopy should be performed with caution in high-risk patients. In addition, completion rates for colonoscopy have been reported to be lower than those for upper gastrointestinal endoscopy, which are nearly 100%. Completion rates for colonoscopy are 97–98% for skilled endoscopists but 80–90% for all endoscopists, and in some cases, insertion to the cecum is time-consuming[9,12].

Completion of colonoscopy requires the endoscope to pass through multiple colonic bends during the insertion of the endoscope from the anus

to the cecum. During passage through the intestinal bends, a load from the endoscopic shaft is applied to the intestinal wall at the intestinal bends, which can lead to complications such as pain and perforation[5,13]. Because the strength of the load from the endoscopic shaft on the intestinal wall depends on the pushing force of the endoscope and the friction between the endoscopic shaft and the colonic mucosal surface, reducing the friction between the endoscopic shaft and the colonic mucosa surface may reduce the load on the intestinal wall during endoscope insertion.

A possible method of reducing friction is the application of lubricant to the colonic mucosal surface. The application of lubricating jelly to the anal region and the endoscopic shaft for lubricating only the anorectal area is currently common in colonoscopy; however, the application of lubricant to the colonic mucosal surface is not standardized. Previous research has reported that the injection of vegetable oil into the intestinal tract may decrease invasiveness during endoscopy insertion, suggesting the

[1]Department of Molecular Gastroenterology and Hepatology, Graduate School of Medical Science, Kyoto Prefectural University of Medicine, Kyoto, Japan. [2]Department of Infectious Diseases, Graduate School of Medical Science, Kyoto Prefectural University of Medicine, Kyoto, Japan. [3]Department of Forensic Medicine, Graduate School of Medical Science, Kyoto Prefectural University of Medicine, Kyoto, Japan. ✉e-mail: ryo-hiro@koto.kpu-m.ac.jp

effectiveness of lubricant application to the colonic mucosal surface[14,15]. However, basic research on lubrication of the endoscopic shaft and colonic mucosa has not been conducted, and the effectiveness of lubricants and the ideal lubricant properties have not been elucidated.

Tribology is the science and engineering of friction and lubrication, and has applications in various medical fields[16–20]. In the boundary and mixed lubrication phases, the friction coefficient decreases as lubricant viscosity and speed increase, and load decreases, but the friction coefficient increases in the hydrodynamic lubrication phase. This behavior of the friction coefficient is conceptually represented as a Stribeck curve[21–24]. As the viscosity of the lubricant increases, the dynamic friction coefficient (DFC) between the endoscopic shaft and colonic mucosa is expected to decrease to a minimum value and then increase. The challenge is to determine the ideal lubricant viscosity that minimizes the DFC between the endoscopic shaft and colonic mucosa.

However, there are significant barriers to determining the ideal lubricant viscosity. It is impossible to measure the DFC between the endoscopic shaft and the human colonic mucosal surface in vivo, and even ex vivo measurements are extremely difficult because it is impractical to procure a sufficient number of colon specimens from surgical patients. The difficulty of measuring DFC has hindered previous investigation of lubrication between the endoscopic shaft and the colonic mucosal surface.

We developed a friction evaluation model using colons obtained from forensic autopsy specimens. At institutions performing forensic autopsies, fresh colons collected within 24 h after death can be stably supplied for research, facilitating the generation of a high-quality and reproducible model[25,26].

Using the friction evaluation model, we first evaluated the influence of the DFC between the endoscopic shaft and colonic mucosal surface on colonoscope insertion. Next, we evaluated the relationship between the DFC and lubricant viscosity. Finally, we evaluated the ease of injecting the lubricant through the endoscope forceps hole. We conducted these evaluations to determine the ideal endoscope lubricant viscosity that best reduces pain and strain on the colon wall during colonoscopy insertion.

## Results

### Relationship between dynamic friction coefficient (DFC) and load on the colonic mucosa during colonoscope insertion

Using various concentrations of hydroxyethyl cellulose (HEC)-based lubricants, we first measured the load exerted by the endoscopic shaft on the colonic mucosa during colonoscopy insertion at a bowel bend (Fig. 1a). The load values were $1600.7 \pm 122.2$ mN with distilled water, $1274.0 \pm 80.0$ mN with 0.1% HEC-based lubricant, $947.3 \pm 104.1$ mN with 0.25% HEC-based lubricant, $659.9 \pm 51.4$ mN with 0.5% HEC-based lubricant, $620.7 \pm 46.2$ mN with 0.9% HEC-based lubricant, and $1143.3 \pm 46.2$ mN with 2.0% HEC-based lubricant (Fig. 1b). Next, we measured the DFC between the colonic mucosa and the endoscopic shaft (Supplementary Fig. S1). The values of the DFC were $0.131 \pm 0.006$ with distilled water, $0.111 \pm 0.016$ with 0.1% HEC-based lubricant, $0.098 \pm 0.025$ with 0.25% HEC-based lubricant, $0.087 \pm 0.022$ with 0.5% HEC-based lubricant, $0.083 \pm 0.012$ with 0.9% HEC-based lubricant, and $0.097 \pm 0.012$ with 2.0% HEC-based lubricant (Fig. 1b). Both the DFC and the load were the lowest when 0.9% HEC-based lubricant was used. In addition, a correlation analysis showed an extremely strong correlation (correlation coefficient of 0.965) between the DFC and the load (Fig. 1c). To minimize the load on the colonic mucosa during colonoscope insertion, a lubricant that minimizes the DFC between the colonic mucosa and the endoscopic shaft is required (Figs. 1d, e).

### Relationship between DFC and insertion time to the cecum

A colonoscopy training simulator was used to evaluate the insertion time to the cecum for different concentrations of HEC-based lubricants. The DFC between the endoscopic shaft and the colonic mucosa (silicone rubber) of the colonoscopy training simulator was also measured. The insertion times to the cecum were >1800 s with distilled water, >1800 s with 0.1% HEC-

based lubricant, $221.6 \pm 117.0$ s with 1.0% HEC-based lubricant, $62.7 \pm 7.5$ s with 3.0% HEC-based lubricant, and $106.4 \pm 21.3$ s with 5.8% HEC-based lubricant. The DFC and insertion time to the cecum with 1.0% HEC-based lubricant were significantly lower and shorter, respectively, than those with water and 0.1% HEC-based lubricant ($p < 0.0001$) (Fig. 2a). The DFC and insertion time to the cecum with 3.0% HEC-based lubricant were significantly lower and shorter, respectively, than those with 1.0% HEC-based lubricant ($p < 0.0001$) (Fig. 2b). The DFC and insertion time to the cecum with 5.8% HEC-based lubricant were significantly higher and longer, respectively, than those with 3.0% HEC-based lubricant ($p < 0.0001$) (Fig. 2c). A lubricant that minimizes the DFC between the colonic mucosa and the endoscopic shaft is required to minimize the insertion time to the cecum.

### Measurement of lubricant viscosity

The viscosities of the carboxymethyl cellulose (CMC), xanthan gum (XG), HEC, sodium alginate (SA), and sodium polyacrylate (SPA)-based lubricants increased with increasing concentration (Supplementary Table S1). The SPA and XG-based lubricants exhibited characteristics of a pseudo-plastic fluid, with viscosity decreasing as the shear rate increases, whereas the CMC, SA, and HEC-based lubricants exhibited characteristics of a Newtonian fluid, with viscosity remaining nearly constant irrespective of the shear rate (Supplementary Fig. S2).

### Relationship between DFC and lubricant viscosity

The DFC between the colonic mucosa and the endoscopic shaft was measured for various concentrations of CMC-, XG-, HEC-, SA-, and SPA-based lubricants. The DFC decreased with increasing concentration of each lubricant. The lubricant concentrations corresponding to the lowest DFCs were 1.0% for CMC, 0.5% for XG, 0.9% for HEC, 0.5% for SA, and 0.08% for SPA. The DFCs with 1.0% CMC, 0.5% XG, 0.9% HEC, 0.5% SA, and 0.08% SPA-based lubricants were $0.097 \pm 0.021$, $0.081 \pm 0.020$, $0.083 \pm 0.013$, $0.092 \pm 0.030$, and $0.085 \pm 0.013$, respectively, and the minimum DFC was almost the same, approximately 0.09, for all lubricants. At concentrations above the concentration corresponding to the minimum DFC, the DFC increased with increasing lubricant concentration (Fig. 3a–e and Supplementary Table S2).

Next, the viscosities of the lubricants were measured (Supplementary Table S1), and the degree of correlation between lubricant viscosities at various shear rates and the DFC was analyzed. The correlation coefficients between the DFC and the lubricant viscosity at shear rates of 0.1, 1, 10, 100, and 1000 1/s were 0.524, 0.697, 0.827, 0.862, and 0.752, respectively (Fig. 4a–e). The DFC showed the strongest correlation with lubricant viscosity at a shear rate of 100 1/s. The viscosities of the 1.0% CMC, 0.5% XG, 0.9% HEC, 0.5% SA, and 0.08% SPA-based lubricants at a shear rate of 100 1/s were 0.053, 0.059, 0.086, 0.063, and 0.031, respectively. Thus, the lubricant with the minimum DFC had a median viscosity of 0.059 (range: 0.031 to 0.086).

### Comparison between conventional gel lubricants and low-friction lubricants

First, the DFC between the colonic mucosa and the endoscopic shaft was evaluated for conventional gel lubricants (Xylocaine Jelly®, CaineZero Jelly®, Through Projelly®, Null Jelly®, K-Y Jelly®, Endolubri-L jelly®, and Endolubri-H jelly®) and low-friction lubricants (1.0% CMC, 0.5% XG, 0.9% HEC, 0.5% SA, and 0.08% SPA-based lubricants). The DFCs with Xylocaine Jelly®, CaineZero Jelly®, Through Projelly®, Null Jelly®, K-Y Jelly®, Endolubri-L jelly®, and Endolubri-H jelly® were $0.131 \pm 0.004$, $0.149 \pm 0.010$, $0.149 \pm 0.012$, $0.178 \pm 0.012$, $0.162 \pm 0.001$, $0.148 \pm 0.007$, and $0.175 \pm 0.011$, respectively. The DFCs of the conventional gel lubricants were significantly higher than those of the low-friction lubricants (Fig. 5a). The viscosities of Xylocaine Jelly®, CaineZero Jelly®, Through Projelly®, Null Jelly®, K-Y Jelly®, Endolubri-L jelly®, and Endolubri-H jelly® at a shear rate of 100 1/s were 1.286, 1.485, 1.377, 2.205, 2.369, 1.388, and 2.150, respectively, and were higher than those of the low-friction lubricants (Fig. 5a and Supplementary Table S2).

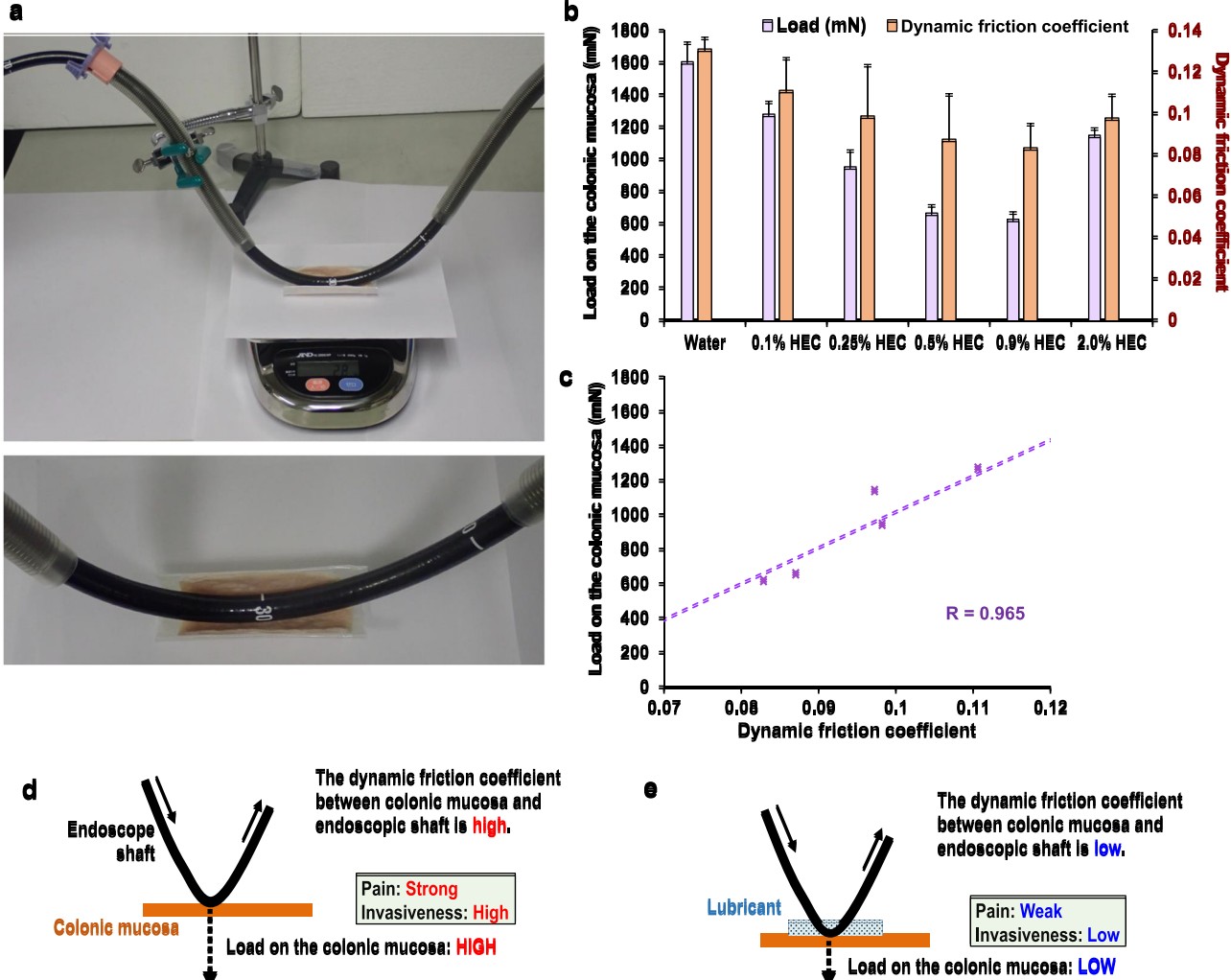

**Fig. 1 | Evaluation of the load on the colonic mucosa during colonoscope insertion. a** Load measurement model. We constructed an ex vivo model to measure the load exerted by the endoscopic shaft on the colonic mucosa during colonoscopy insertion at a bowel bend. Specifically, the colon autopsy specimen was fixed to the weighing pan of an electronic analytical scale, and the bowel bend was reproduced with overtubes. **b** Measurement of the load on the colonic mucosa using hydroxyethyl cellulose (HEC)-based lubricants. The results obtained were expressed in terms of the mean ± standard error. **c, d, e** Relationship between the dynamic friction coefficient (DFC) and the load. A Pearson correlation analysis between the DFC and the load was performed (**c**). To minimize the load on the colonic mucosa during colonoscope insertion, a lubricant that minimizes the DFC between the colonic mucosa and the endoscopic shaft was required (**d, e**).

Next, the DFC between the skin surface and the endoscopic shaft was evaluated for the conventional gel lubricants and low-friction lubricants. The DFCs with Xylocaine Jelly®, CaineZero Jelly®, Through Projelly®, Null Jelly®, K-Y Jelly®, Endolubri-L jelly®, and Endolubri-H jelly® were 0.104 ± 0.000, 0.121 ± 0.002, 0.118 ± 0.002, 0.145 ± 0.003, 0.131 ± 0.000, 0.123 ± 0.002, and 0.138 ± 0.002, respectively. The DFCs with 1.0% CMC, 0.5% XG, 0.9% HEC, 0.5% SA, and 0.08% SPA-based lubricants were 0.396 ± 0.022, 0.539 ± 0.050, 0.478 ± 0.010, 0.590 ± 0.010, and 0.406 ± 0.036, respectively. The DFCs with the conventional gel lubricants were significantly lower than those with the low-friction lubricants (Fig. 5b and Supplementary Table S3).

The viscosities of the conventional gel lubricants are suitable for minimizing the DFC between the skin surface and the endoscopic shaft but are too high to minimize the DFC between the colonic mucosa and the endoscopic shaft.

**Relationship between injection pressure and lubricant viscosity**
For ideal lubrication of the endoscopic shaft and the colonic mucosa, low-friction lubricants should be injected through the forceps hole of the endoscope. The injection pressure, an index of the ease of injection, was measured

for each lubricant, and a correlation analysis between injection pressure and lubricant viscosity was performed. The correlation coefficients between injection pressure at an injection speed of 0.5 mL/s and lubricant viscosity at shear rates of 0.1, 1, 10, 100, and 1000 1/s were 0.184, 0.253, 0.544, 0.916, and 0.979, respectively. For all injection speed conditions (0.2–0.5 mL/s), the injection pressure showed the strongest correlation with lubricant viscosity at a shear rate of 1000 1/s (Fig. 6a–e). The results of the correlation analysis indicated that among lubricants with the same DFC, pseudoplastic fluids such as SPA- and XG-based lubricants are easier to inject than Newtonian fluids, such as CMC-, SA-, and HEC-based lubricants (Fig. 6f).

The injection pressures of the conventional gel lubricants (Xylocaine Jelly®, CaineZero Jelly®, Through Projelly®, Null Jelly®, K-Y Jelly®, Endolubri-L jelly®, and Endolubri-H jelly®) exceeded 80 psi, the measurement limit. The injection pressures of the low-friction lubricants (1.0% CMC, 0.5% XG, 0.9% HEC, 0.5% SA, and 0.08% SPA) at an injection speed of 0.5 mL/s were 43.90 ± 0.14, 19.19 ± 0.36, 51.20 ± 2.40, 35.78 ± 1.68, and 17.74 ± 2.40, respectively. The pseudoplastic fluids (0.08% SPA- and 0.5% XG-based lubricants) had significantly lower injection pressures and lower viscosities at a shear rate of $10^3$ 1/s than the Newtonian fluids (1.0% CMC-, 0.5% SA-, and 0.9% HEC-based lubricants) ($P < 0.001$ for all) (Figs. 6f, g).

**Fig. 2 | Relationship between the dynamic friction coefficient (DFC) and insertion time to the cecum.** A colonoscopy training simulator was used to measure insert time to the cecum. Further, the DFC between the endoscopic shaft and the colonic mucosa (silicone rubber) of the colonoscopy training simulator was measured. Distilled water (**a**), 0.1% (**a**), 1.0% (**a**, **b**), 3.0% (**b**, **c**), and 5.8% (**c**) hydroxyethyl cellulose (HEC)-based lubricants were evaluated. Data are presented in three panels to clarify comparisons (**a**-**c**) and, expressed in terms of the mean ± standard error of at least three independent experiments.

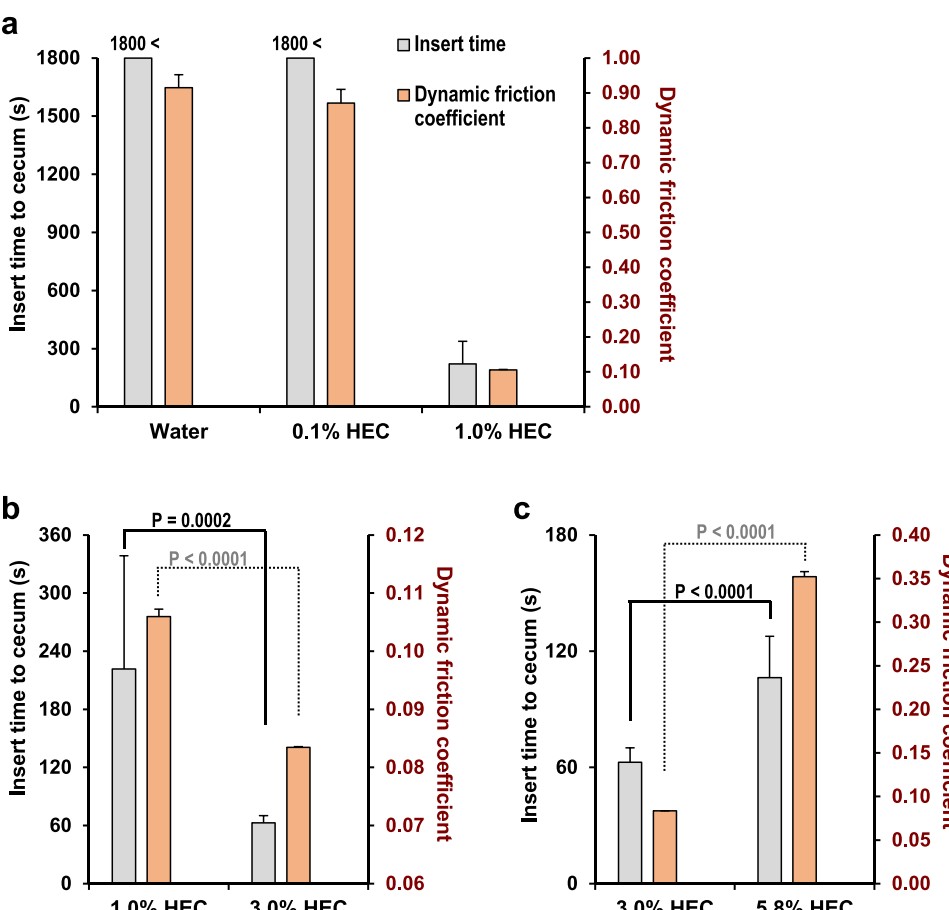

## Discussion

Colonoscopy is fairly highly invasive, with a high incidence of complications such as pain and perforation during endoscopy insertion[5–11]. Reducing the DFC between the endoscopic shaft and the colonic mucosa surface may reduce the invasiveness of colonoscopy. However, investigations on lubrication of the endoscopic shaft and colonic mucosa surface have not been conducted previously, and no lubricant has been developed to reduce the DFC between the endoscopic shaft and the colonic mucosa surface. The method and theory of friction evaluation to identify the optimal viscosity properties of lubricants have already been established[21–24]. However, measuring the DFC between the endoscopic shaft and the colonic mucosa surface is extremely difficult both in vivo and ex vivo, which has been an obstacle to its investigation. To overcome this obstacle, we generated a friction measurement model using colons obtained from forensic autopsy specimens, and we obtained accurate and reproducible measurements.

The evaluation of HEC-based lubricants showed that the lubricant that minimized the DFC between the endoscopic shaft and the colonic mucosa could minimize the load on the colonic mucosa during colonoscope insertion and could reduce the endoscopy insertion time to the cecum. The use of an ideal lubricant that minimizes the DFC may shorten the total colonoscopy time, alleviate pain during colonoscopy, and reduce the invasiveness of colonoscopy.

The relationship between the lubricant viscosity and the DFC between the endoscopic shaft and the colonic mucosa was evaluated, and the viscosity of the lubricant with the lowest DFC (i.e., an ideal lubricant) was determined. The strong correlation between the lubricant viscosity and the DFC can be easily inferred from the Stribeck curve. Nevertheless, because the candidate lubricant materials include both pseudoplastic fluids (non-Newtonian fluids) with shear rate-dependent viscosity and Newtonian fluids with shear rate-independent viscosity, identifying a shear rate compatible with the situations of common colonoscopy insertion (i.e.,

colonoscope insertion at approximately 1 cm/s) was important. Therefore, a correlation analysis between the DFC and the lubricant viscosity at various shear rates was performed. The highest correlation was found between the DFC and the viscosity at a shear rate of 100 1/s. Thus, the DFC was determined by the lubricant viscosity at a shear rate of 100 1/s irrespective of the type of lubricant material, and the DFC can be predicted by focusing on the viscosity at a shear rate of 100 1/s.

In this study, 1.0% CMC, 0.5% XG, 0.9% HEC, 0.5% SA, and 0.08% SPA were the lubricants with the lowest DFCs (i.e., the low-friction lubricants), and their viscosities at a shear rate of 100 1/s were in the range of 0.031–0.086 (median 0.059). The lowest DFC was almost the same, approximately 0.09, irrespective of the chemical composition of the lubricant, and was observed to depend on the viscosity rather than the chemical composition. These results suggest that the DFCs of lubricants with other chemical components can be minimized by adjusting their concentration to achieve a viscosity in the range of 0.031–0.086 at a shear rate of 100 1/s.

Conventional gel lubricants (e.g., Xylocaine Jelly®, CaineZero Jelly®, Through Projelly®, Null Jelly®, K-Y Jelly®, Endolubri-L jelly®, and Endolubri-H jelly®), which are commonly used in colonoscopy, were developed to lubricate the skin near the anus and the endoscopic shaft. In this study, the DFCs between the endoscopic shaft and the human skin surface were significantly lower with the conventional gel lubricants than with the low-friction lubricants. In contrast, the viscosities of the conventional gel lubricants were considerably higher (at a shear rate of 100 1/s) than those of the low-friction lubricants, and the DFC between the endoscopic shaft and the colonic mucosa surface was significantly higher with the conventional gel lubricants than with the low-friction lubricants. These results indicate that conventional gel lubricants have viscosities that are suitable for lubrication between the skin surface and the endoscopic shaft, although their viscosities are too high for lubrication between the

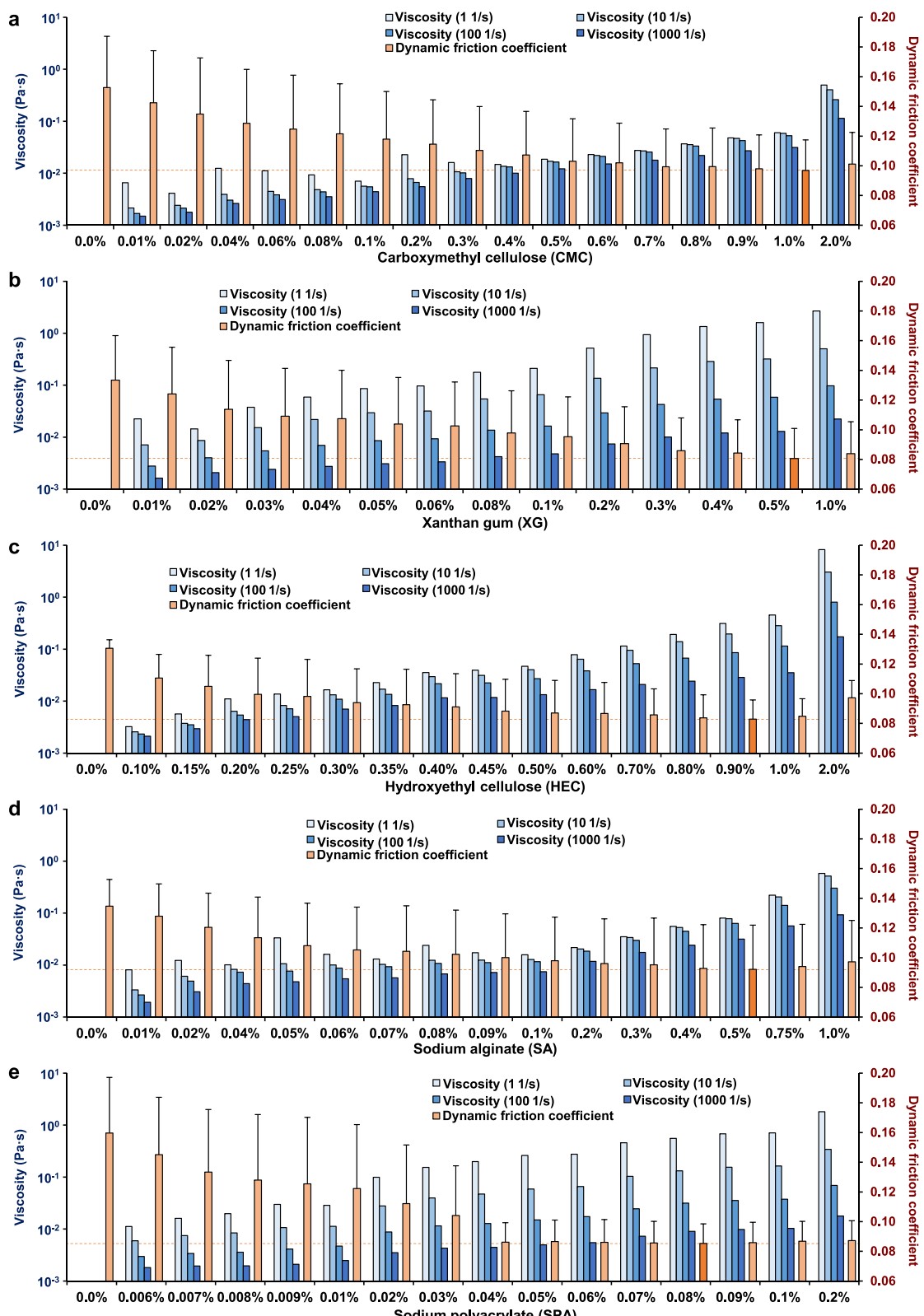

**Fig. 3 | Measurement of the lubricant viscosity and the dynamic friction coefficient (DFC) between the colonic mucosa and the endoscopic shaft.** Viscosity and the DFC of carboxymethyl cellulose (CMC)-based lubricant (**a**), xanthan gum (XG)-based lubricant (**b**), hydroxyethyl cellulose (HEC)-based lubricant (**c**), sodium alginate (SA)-based lubricant (**d**), and sodium polyacrylate (SPA)-based lubricant (**e**) at each concentration were measured. The minimum DFC for each lubricant is indicated by the dotted red line. Data are expressed in terms of the mean ± standard error of at least three independent experiments.

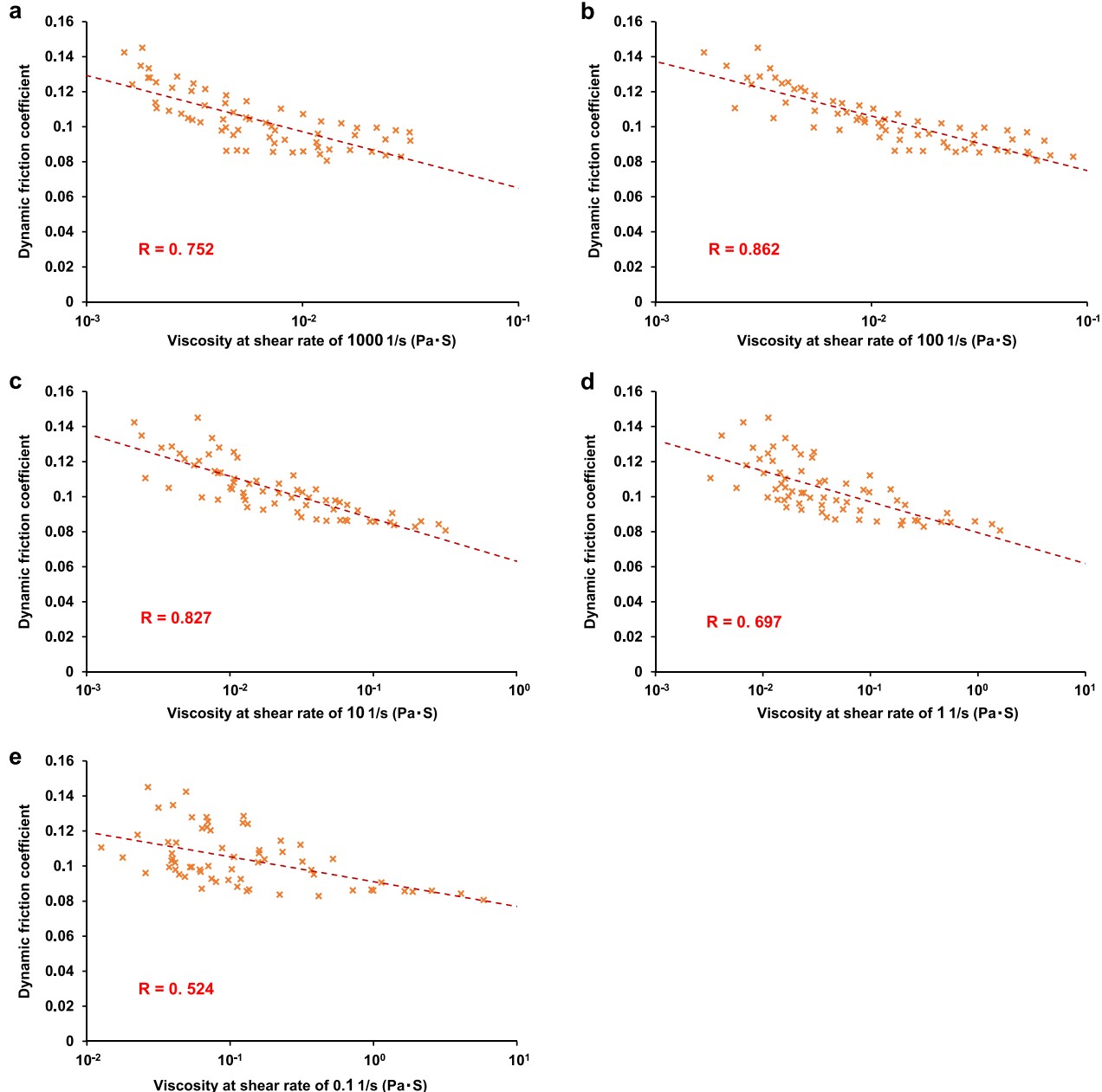

**Fig. 4 | Scatter plot of the dynamic friction coefficient (DFC) versus the lubricant viscosity.** The Pearson correlations between the dynamic friction coefficient (DFC) and the lubricant viscosity at shear rates of 1000 1/s (**a**), 100 1/s (**b**), 10 1/s (**c**), 1 1/s (**d**), and 0.1 1/s (**e**) were analyzed.

endoscopic shaft and the colonic mucosa. In colonoscopy, lubrication of the anorectal area and lubrication of the colonic mucosa need to be addressed separately, and different lubricants suitable for each lubrication should be used. Specifically, for lubrication of the anorectal area, the conventional method of directly applying a conventional gel lubricant to the anorectal area and endoscopic shaft before starting endoscope insertion is optimal. For lubrication of the colonic mucosa, a low-friction lubricant should be applied to the colonic mucosa surface surrounding the endoscope tip by injecting the low-friction lubricant through the endoscope forceps hole during endoscope insertion. Because the procedure for injecting water into the intestinal tract through the endoscope forceps hole during endoscope insertion has been used previously for intestinal cleansing and the water immersion colonoscopy insertion technique, lubrication of the colonic mucosa, which is a similar procedure, is easy to perform[27,28].

Finally, as low-friction lubricants are intended to be injected through the endoscope forceps hole, the injection pressure of lubricants was

evaluated. A correlation analysis showed that the highest correlation between the injection pressure and the lubricant viscosity occurred at a shear rate of 1000 1/s. Thus, the injection pressure was determined by the lubricant viscosity at a shear rate of 1000 1/s irrespective of the type of lubricant material, and the injection pressure can be predicted by focusing on the viscosity at 1000 1/s. Because the injection pressure and DFC are determined by the viscosity at different shear rates, a comparison of Newtonian-fluid and pseudoplastic-fluid lubricants with similar DFCs (i.e., similar viscosities at a shear rate of 100 1/s) revealed that the pseudoplastic-fluid lubricants had lower injection pressures (i.e., lower viscosities at a shear rate of 1000 1/s). Evaluation of the injection pressure at an injection speed of 0.5 mL/s, which is close to the actual injection speed during colonoscopy, indicated that the injection pressures of the pseudoplastic-fluid lubricants (e.g., 0.08% SPA- and 0.5% XG-based lubricants) were significantly lower than those of the Newtonian-fluid lubricants (e.g., 1.0% CMC-, 0.5% SA-, and 0.9% HEC-based lubricants). These findings indicate that pseudoplastic-fluid

**Fig. 5 | Comparison between conventional gel lubricants and low-friction lubricants.**
**a** Viscosity at a shear rate of 100 1/s, and the dynamic friction coefficient (DFC) between the colonic mucosa and the endoscopic shaft. **b** DFC between the skin surface and the endoscopic shaft. Data are expressed in terms of the mean ± standard error of at least three independent experiments. CMC carboxymethyl cellulose, XG xanthan gum, HEC hydroxyethyl cellulose, SA sodium alginate, SPA sodium polyacrylate.

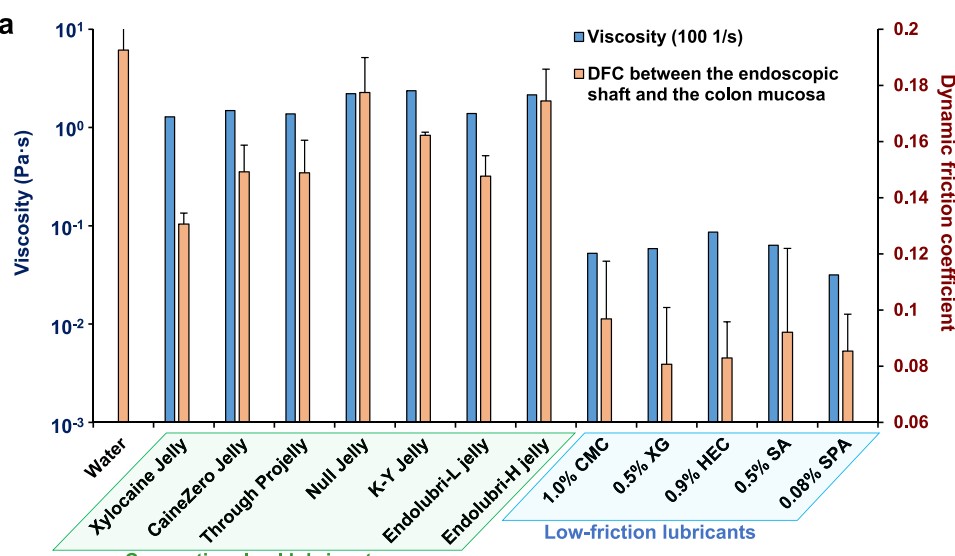

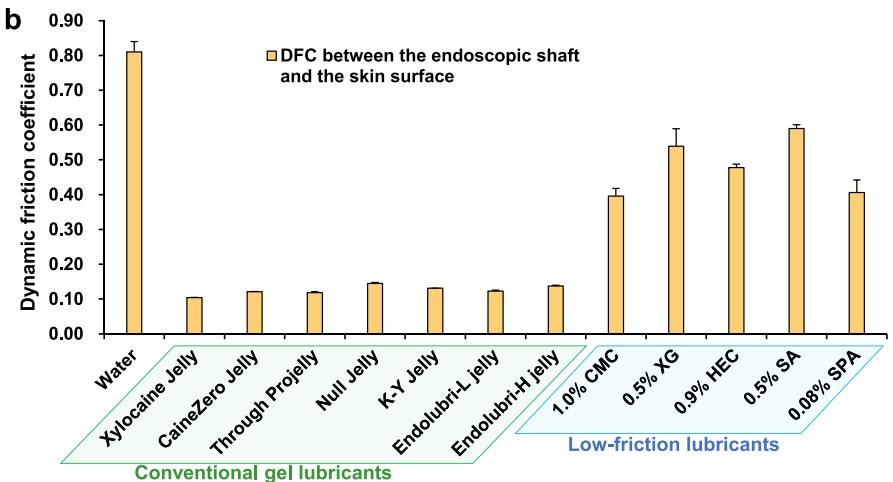

lubricants such as 0.08% SPA- and 0.5% XG-based lubricants are low-friction lubricants that can be considered ideal. These ideal low-friction lubricants can be easily injected through the endoscope forceps hole with a syringe or an endoscopic spray tube and can also be injected through the endoscope water delivery port. Therefore, these ideal low-friction lubricants can also be used as an alternative to flushing water during examination for removing intestinal residues and securing the field of view. Consequently, these ideal low-friction lubricants can be flexibly adapted to various uses, thus improving the quality of colonoscopy without changing the current examination style.

Three limitations of this study should be noted. First, the colons were obtained from forensic autopsy specimens within 24 h of death and used for friction measurement without any additional treatment. In other words, the colon mucosa with mucus still attached was used to evaluate friction. Thus, in this study, the friction coefficient was measured under the coexistence of mucus and lubricant on the colonic mucosa surface (i.e., the same condition as actual colonoscopy). However, the interaction between mucus and lubricants is unclear and requires further research. In addition, it cannot be assured that these colons are in exactly the same condition as the living colon. Nevertheless, as it is impossible to measure the DFC between the endoscopic shaft and the human colonic mucosal surface in vivo, the friction measurement model provides the most accurate and reproducible analysis possible at this stage. Second, pure water was used as the solvent for lubricants rather than an electrolyte

liquid, such as saline, because cations such as sodium and calcium ions could change the lubricant viscosity[29]. Even if the lubricant viscosity were to change due to cations present during use in vivo and formulation, it is possible to accurately predict the DFC and injection pressure at a given point by measuring the lubricant viscosity after the change. In addition, during actual colonoscopy, tepid water containing no salts is often injected into the colon to clean it and secure the visual field, which is another reason why pure water was used as the solvent for lubricants. Third, SPA might be excluded from the candidate lubricant materials because SPA, an absorbent polymer, has been reported as a possible cause of intestinal obstruction[30,31].

In conclusion, based on the principles of tribology, we formulated the friction measurement model that reproduced almost the same situation as a living body and identified the viscosity characteristics of an ideal lubricant that minimizes the DFC between the endoscopic shaft and colonic mucosa. Introduction of this ideal lubricant to colonoscopy is easy and will alleviate patient pain during colonoscopy insertion, reduce the invasiveness of colonoscopy, and shorten the total colonoscopy time.

## Methods
### Preparation of target lubricants
Target lubricants were prepared by dissolving CMC (FUJIFILM Wako Pure Chemical Corporation, Osaka, Japan), XG (Tokyo Chemical Industry, Tokyo, Japan), HEC (Tokyo Chemical Industry), SA (Nacalai Tesque,

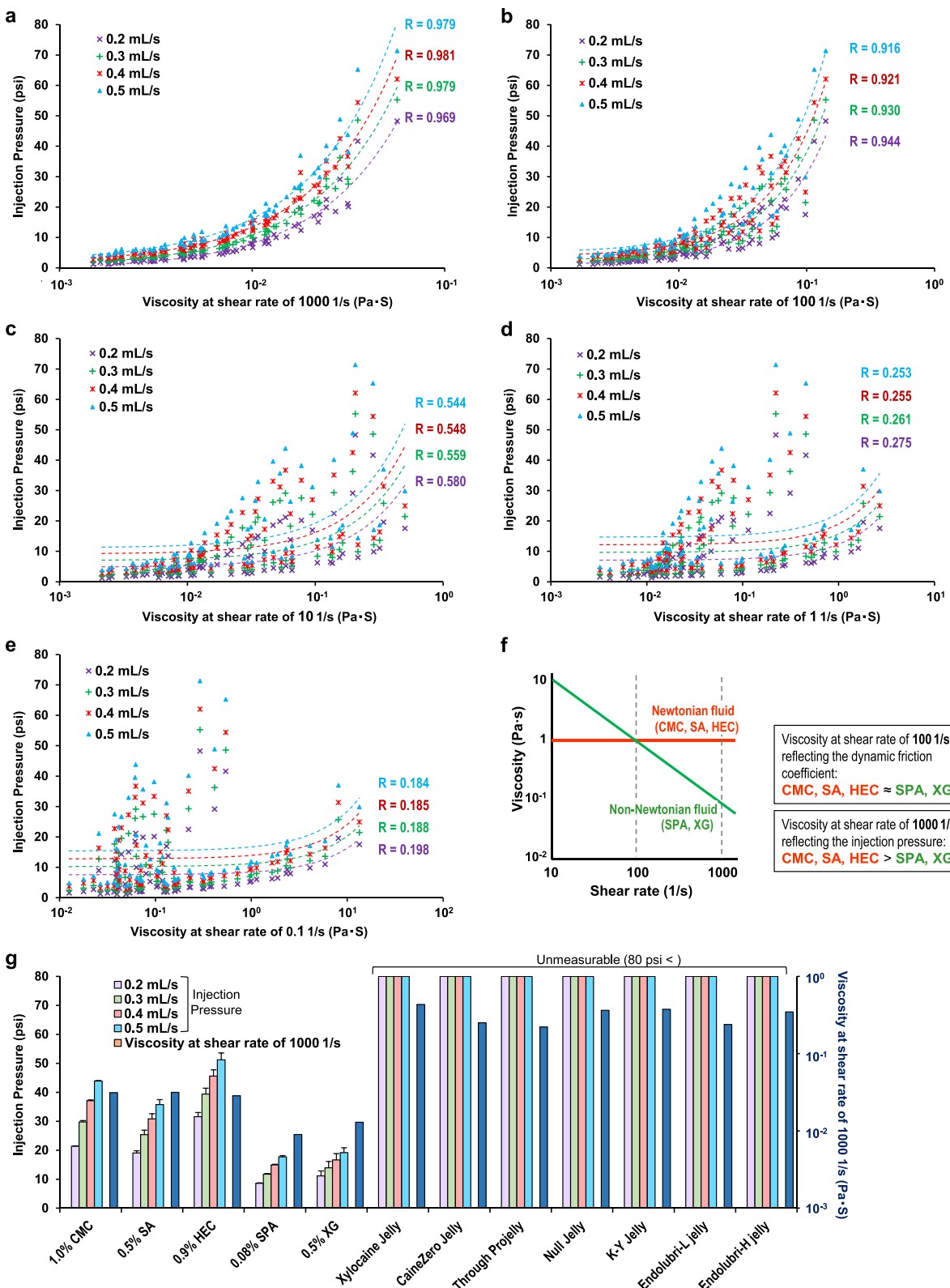

**Fig. 6 | Evaluation of injection pressure.** The Pearson correlations between the injection pressure and the lubricant viscosity at shear rates of 1000 (**a**), 100 (**b**), 10 (**c**), 1 (**d**), and 0.1 1/s (**e**) were analyzed. **f** Among lubricants with the same dynamic friction coefficient, pseudoplastic fluids such as sodium polyacrylate (SPA)-, xanthan gum (XG)-, and hydroxyethyl cellulose (HEC)-based lubricants are easier to inject than Newtonian fluids such as carboxymethyl cellulose (CMC)- and sodium alginate (SA)-based lubricants. **g** Injection pressures of conventional gel lubricants and low-friction lubricants at each injection rate. Data are expressed in terms of the mean ± standard error of at least three independent experiments.

Kyoto, Japan), or SPA (Toagosei, Tokyo, Japan) in pure water. CMC-based lubricants were prepared at concentrations of 0.01–2.0% (w/v), XG-based lubricants at concentrations of 0.01–1.0% (w/v), HEC-based lubricants at concentrations of 0.1–2.0% (w/v), SA-based lubricants at concentrations of 0.01–1.0% (w/v), and SPA-based lubricants at concentrations of 0.006–0.2% (w/v). Because, during actual colonoscopy, tepid water containing no salts is often injected into the colon to clean it and secure the visual field, pure water was used as the solvent for the lubricants in this study. Moreover, because the measured DFC is affected by the lubricant viscosity, not the lubricant concentration, the conditions for the target lubricants were set based on viscosity values. Although lubricants with different solutes had different adjusted concentrations, their viscosity range was generally constant.

In addition, conventional gel lubricants Xylocaine Jelly® (Sandoz Pharma, Tokyo, Japan), CaineZero Jelly® (Shionogi Pharma, Osaka, Japan), Through Projelly® (Kaigen Pharma, Osaka, Japan), Null Jelly® (Nichi-Iko Pharmaceutical, Toyama, Japan), K-Y Jelly® (Reckitt Benckiser Healthcare, Berkshire, UK), Endolubri-L jelly® (Saraya, Osaka, Japan), and Endolubri-H jelly® (Saraya) were evaluated in this study.

## Rheological analysis
The viscosities of the lubricants were measured using the Discovery HR-1 rheometer (TA Instruments, Surrey, UK) with a 60 mm cone plate geometry at 25 °C and 37 °C[29,32,33]. A Peltier plate controlled the temperature. Steady-flow viscosity was measured in flow sweep mode (steady-flow measurement). The viscosity was measured for a range of shear rates (0.01, 0.016, 0.025, 0.04, 0.063, 0.1, 0.16, 0.25, 0.4, 0.63, 1.0, 1.6, 2.5, 4.0, 6.3, 10, 16, 25, 40, 63, 100, 160, 250, 400, and 1000 1/s). The viscosity (Pa s) was calculated from the shear stress (Pa) and shear rate (1/s) using TRIOS ver. 4.4.0.41651 (TA Instruments).

## Evaluation of insertion time to the cecum using a colonoscopy training simulator
A colonoscopy training simulator (MW24 NKS Colonoscope Training Simulator, Kyoto Kagaku, Kyoto, Japan) was used to evaluate the insertion time to the cecum for different concentrations of HEC-based lubricants. A colonoscope (CF-Q260AI, Olympus, Tokyo, Japan) was inserted into the colonoscope training simulator, which was sufficiently coated with 200 mL of each lubricant, and the insertion time from the anus to the cecum was measured. Distilled water as well as 0.1%, 1.0%, 3.0%, and 5.8% HEC-based lubricants were used in this evaluation. Four endoscopy specialists (R.H., N.W., H.M., and T.Y.), with experience of conducting more than 1,000 colonoscopies each, performed the colonoscopies. The lubricant used during each colonoscopy was randomized, and the endoscopists were blinded to the lubricant used. Three colonoscopy insertions per lubricant were performed.

## Collection of human colon and skin from autopsy specimens
Human colon and skin samples were collected from forensic autopsy specimens obtained from the Department of Forensic Medicine of the Kyoto Prefectural University of Medicine. Total colon and abdominal skin autopsy specimens from individuals aged 20–70 y were obtained within 24 h of death. Considerably damaged or necrotic colon and skin specimens were excluded. The specimens obtained were immediately stored at −80 °C. To ensure uniform conditions, all frozen specimens were thawed immediately before the analysis procedure was conducted[25,26].

## Evaluation of the load on the colonic mucosa during colonoscope insertion
An evaluation model was constructed to measure the load exerted by the endoscopic shaft on the colonic mucosa during colonoscopy insertion at a bowel bend (Fig. 1a). A colon autopsy specimen was fixed to the weighing pan of an electronic analytical scale (HL-3000LWP, A&D, Tokyo, Japan), and the bowel bend was reproduced with overtubes (TOP, Tokyo, Japan). After 0.5 mL of lubricant was evenly applied to the mucosa of the colon autopsy specimen, the endoscopic shaft was brought into contact with the colonic mucosa, and the colonoscope (CF-Q260AI) was inserted 10 cm at a rate of 1 cm/s. The maximum load measured during the endoscope insertion was defined as the load on the colonic mucosa. Distilled water as well as 0.1%, 0.25%, 0.5%, 0.9%, and 2.0% HEC-based lubricants were used, and three independent measurements were performed for each lubricant. The results obtained were expressed in terms of the mean ± standard error.

## Model for measuring the dynamic friction coefficient (DFC) between colonic mucosa and endoscopic shaft
The DFC was measured using TRIBOGEAR TYPE 38 (Shinto Scientific, Tokyo, Japan) and calculated using Tribosoft ver. 6.26 (Shinto Scientific). A colon autopsy specimen was fixed to a moving table with a certain tension applied, and the DFC between the colonic mucosa and the endoscopic shaft was measured by attaching the endoscopic shaft portion to the measurement fixture (Supplementary Fig. S1). The material of the endoscopic shaft was fluorine resin.

After 0.5 mL of lubricant was evenly applied to the mucosa of the colon autopsy specimen, the endoscopic shaft portion was brought into contact with the colonic mucosa and loaded with a 100-g weight. The moving table was set to move a distance of 50 mm at a speed of 1.0 cm/s and a reciprocation frequency of 10. The average of the DFC values measured during the 10 round trips of the moving table was considered as the measured value. Colon autopsy specimens obtained from three different individuals were used for this measurement. Three independent measurements were performed for each lubricant, and the results obtained were expressed in terms of the mean ± standard error.

## Measurement of the DFC between the skin and endoscopic shaft and between the colonic mucosa (silicone rubber) of the colonoscopy training simulator and endoscopic shaft
The DFC was measured using skin autopsy specimens or the colonic mucosa (silicone rubber) of the colonoscopy training simulator rather than colon autopsy specimens. The DFC between the skin and the endoscopic shaft and between the colonic mucosa of the colonoscopy training simulator and the endoscopic shaft was measured in a manner similar to that described above.

## Measurement of injection pressure
The injection pressure of each lubricant was evaluated by connecting an endoscopic spray tube (W2825, TOP, Tokyo, Japan), a digital pressure gauge (BN-PGD60PL-F1, Nihon Seiki, Osaka, Japan), and a syringe containing a solution to a three-way stopcock (TS-TL1K, TERUMO, Tokyo, Japan)[29,32]. The digital pressure gauge was connected to the three-way stopcock by an infusion extension tube. The injection pressure magnitude was measured when the pressure value was stable for 3 s, using a syringe pump (Legato 100, KD Scientific Inc, Holliston, USA) for which the injection rate and syringe type can be set. A 20-mL syringe (TERUMO) was used, and injection rates of 0.2, 0.3, 0.4, and 0.5 mL/s were considered. Three independent measurements were performed for each lubricant, and the results were expressed in terms of the mean ± standard error.

## Ethical considerations
The study protocol, including sample collection procedures, was reviewed and approved by the Institutional Review Board of the Kyoto Prefectural University of Medicine (ERB-C-2573).

## Statistical analysis
The data obtained were analyzed using GraphPad Prism 7 (GraphPad Inc., La Jolla, CA, USA). The Pearson's correlation coefficient was used to assess the degrees of correlations between the logarithm of viscosity and the DFC, the logarithm of viscosity and the injection pressure, and the DFC and the

load on the colonic mucosa. Continuous variables were evaluated using Student's $t$-test. All reported $p$-values were two-sided, and values with $p < 0.05$ were considered significant.

## Reporting summary

Further information on research design is available in the Nature Portfolio Reporting Summary linked to this article.

## Data availability

All data included in this study are available from the corresponding author on request.

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

## Acknowledgements

The authors thank Editage (www.editage.com) for English language editing. We also thank all members of the Department of Molecular Gastroenterology and Hepatology, Kyoto Prefectural University of Medicine, for assistance with the study. This work was supported by TERUMO LIFE SCIENCE FOUNDATION and "The Translational Research program; Strategic PRomotion for practical application of INnovative medical Technology (TR-SPRINT)."

## Author contributions

Study concept and design: R.H. Acquisition of data: N.W., R.H., H.I., K.Y., H.M., T.Y., R.B., K.I., O.D., N.Y., T.N., and Y.I. Data analysis and interpretation: N.W. and R.H. Drafting of the manuscript: N.W. and

R.H. Critical revision of the manuscript for intellectual content: R.H. Statistical analysis: N.W. and R.H. Secured funding: R.H. Administrative/technical/material support: R.H., H.I., T.N., and Y.I. Study supervision: R.H.

## Competing interests

The authors declare no competing interests.

## Additional information

**Supplementary information** The online version contains Supplementary Material available at https://doi.org/10.1038/s44172-024-00177-5.

