## [Peer Review File · Communications Engineering]

Reviewers' comments:

Reviewer #1 (Remarks to the Author):

The paper aims to find out the ideal lubricant viscosity to decrease colonoscopy invasiveness and total colonoscopy time. Overall, I think this is an interesting study that should be published in this journal. However, I have a list of items that need to be addressed in a revision. Those items are outlined below:

1. Indeed, the lubricant viscosity can be affected by the ions in the solvent, but it depends on the ion strength. Thus I prefer to use saline as the solvent cause it's more physiologically related.
2. The authors claim that they use a newly constructed friction measurement model to identify the viscosity characteristics of an ideal lubricant. However, I think the characterization methods are the traditional ones. Please give more evidence to support this point.
3. The concentrations of target lubricants are quite different, e.g., CMC 0.01–2.0% and SPA 0.006–0.2%.
4. What is the pH of the target lubricant? The effect of pH should be taken into consideration.
5. The precision of DFGs presented in Table S2 is 0.0001. Can the sensors detect such a high level?
6. What is the material of the endoscopic shaft? The tribological behavior is highly related to the tribo-pair.
7. Figure S1 and Figure 5C make no sense since no valuable information can be obtained. The authors should either use their own data to plot or just cited a reference.
8. The legend of Figure 6 should be “mL”.

Reviewer #2 (Remarks to the Author):

This paper provides a comprehensive exploration of the impact of various lubricants on the friction coefficient during colonoscopic procedures. The authors innovatively developed a DFC measurement model using colons from forensic autopsy specimens. This model not only reveals the optimal viscosity of lubricants that can reduce the DFC but also further illustrates how these lubricants alleviate the burden on the colonic mucosa during colonoscope insertion. At specific shear rates, there's a notable correlation between the viscosity of the lubricant and the DFC, which aids in shortening the insertion time of the colonoscope from the anus to the cecum. These findings offer strong evidence for reconsidering the choice of lubricants in colonoscopies to enhance patient comfort and surgical efficiency. However, the authors might consider the following suggestions to further refine the paper

(1) While the paper considers the impact of various lubricants on the colonic friction coefficient, there seems to be no in-depth discussion of the relationship between lubricants and the natural mucus

of the colon. The colon itself secretes a certain amount of mucus to lubricate and protect its surface. When lubricants mix with this mucus, they might interact, altering the overall viscosity and lubricity. It is suggested to further study the frictional performance of lubricants in simulated colonic mucus to more accurately mimic the real colonic environment.

(2) The paper uses water as a control group. However, considering that water might dilute the natural mucus of the colon, it is recommended to use a liquid closer to the natural state of the colon (such as saline or simulated colonic fluid) as a control group, or explicitly state this limitation and improve it in subsequent research.

(3) The paper utilized a colonoscope of a specific size and model to measure the DFC. However, in practice, colonoscopes of different models and sizes might exert varying pressures on the colon, leading to differences in the friction coefficient. It is recommended that the authors conduct the DFC tests across a range of colonoscope types and sizes to evaluate the influence of instrument size and design on lubrication. This would lend greater universality to the results.

Responses to comments from the reviewers

=====

Reviewer #1:

The paper aims to find out the ideal lubricant viscosity to decrease colonoscopy invasiveness and total colonoscopy time. Overall, I think this is an interesting study that should be published in this journal. However, I have a list of items that need to be addressed in a revision. Those items are outlined below:

1. Indeed, the lubricant viscosity can be affected by the ions in the solvent, but it depends on the ion strength. Thus, I prefer to use saline as the solvent cause it's more physiologically related.

Response: We appreciate your helpful comments. As the reviewer commented, saline may be preferable from a physiological perspective. However, during actual colonoscopy, tepid water rather than saline is often injected into the colon to clean it and secure the visual field. Because it our developed lubricants are expected to be used in place of the tepid water in actual colonoscopy, pure water containing no salts was used as the solvent for the lubricants in this study. This explanation has been added to the methods and discussion sections (study limitations).

2. The authors claim that they use a newly constructed friction measurement model to identify the viscosity characteristics of an ideal lubricant. However, I think the characterization methods are the traditional ones. Please give more evidence to support this point.

Response: Thank you for your helpful suggestion. As you pointed out, the method and theory of friction evaluation to identify the viscosity properties of lubricants in this study is based on previous studies. Therefore, several references related to the previous studies have been added to the discussion section. The novelty of this study is the achievement of friction measurement under almost the same situation as a living body. Specifically, the evaluation of friction between the colonic mucosal surface and the endoscope was achieved using an *ex vivo* model generated from the colons obtained from forensic autopsy specimens within 24 h of death. The discussion section has been revised to clearly explain this point.

3. The concentrations of target lubricants are quite different, e.g., CMC 0.01–2.0% and SPA 0.006–0.2%.

Response: We appreciate your helpful comments. Because the measured DFC is affected by the lubricant viscosity, not the lubricant concentration, the conditions for the target lubricants were set based on viscosity values. Specifically, each target lubricant was adjusted in several concentrations to achieve viscosities in the range of 0.001–1.0. For example, because SPA is viscous even at very low concentrations, the concentration of the SPA-based target lubricants was low. Therefore, although lubricants with different solutes have different adjusted concentrations, their viscosity range is generally constant. The above explanation has been added to the methods section.

4. What is the pH of the target lubricant? The effect of pH should be taken into consideration.

Response: We appreciate this important comment. As you pointed out, polysaccharide thickeners have an optimum pH range, and their viscosity may change depending on their pH. The pH of the target lubricant is generally within the range of 5.5–8.5, which has been confirmed to be within the optimum pH range.

5. The precision of DFGs presented in Table S2 is 0.0001. Can the sensors detect such a high level?

Response: Thank you for pointing out this error. The measurement precision of our friction tester is 0.001. Therefore, we have changed the description of the data in Supplementary Tables S2 and S3.

6. What is the material of the endoscopic shaft? The tribological behavior is highly related to the tribo-pair.

Response: We appreciate this helpful query. The material of the endoscopic shaft was fluorine resin. A description of the material of the endoscopic shaft has been added to the methods section.

7. Figure S1 and Figure 5C make no sense since no valuable information can be obtained. The authors should either use their own data to plot or just cited a reference.

Response: Thank you for the helpful suggestion. As indicated, Figures S1 and 5C do not contain any new data. Figures S1 and 5C have been removed, and new references have been added as appropriate, because it is possible to discuss research data without these figures.

8. The legend of Figure 6 should be “mL”.

Response: Thank you for pointing out this error. The legend of Figure 6 has been corrected from "ml" to "mL".

=====

Reviewer #2:

This paper provides a comprehensive exploration of the impact of various lubricants on the friction coefficient during colonoscopic procedures. The authors innovatively developed a DFC measurement model using colons from forensic autopsy specimens. This model not only reveals the optimal viscosity of lubricants that can reduce the DFC but also further illustrates how these lubricants alleviate the burden on the colonic mucosa during colonoscope insertion. At specific shear rates, there's a notable correlation between the viscosity of the lubricant and the DFC, which aids in shortening the insertion time of the colonoscope from the anus to the cecum. These findings offer strong evidence for reconsidering the choice of lubricants in colonoscopies to enhance patient comfort and surgical efficiency. However, the authors might consider the following suggestions to further refine the paper.

1. While the paper considers the impact of various lubricants on the colonic friction coefficient, there seems to be no in-depth discussion of the relationship between lubricants and the natural mucus of the colon. The colon itself secretes a certain amount of mucus to lubricate and protect its surface. When lubricants mix with this mucus, they might interact, altering the overall viscosity and lubricity. It is suggested to further study the frictional performance of lubricants in simulated colonic mucus to more accurately mimic the real colonic environment.

Response: Thank you for your helpful suggestion. We recognize the reviewer's concern at the outset of the study. In this study, the colons were obtained from forensic autopsy specimens within 24 h of death and used for research without any additional treatment. In other words, the colon mucosa with mucus still attached was used to evaluate friction. Therefore, in this study, the friction coefficient was measured under the coexistence of mucus and lubricant on the colonic mucosa surface (i.e. the same situation as actual colonoscopy). However, the interaction between mucus and lubricants is not completely clarified, and further research is required. This explanation has been added to the discussion section (study limitations).

As an additional preliminary analysis, we measured the friction coefficient under the condition that pure water or lubricant was applied to the colonic mucosa after mucus was thoroughly washed and removed, and under the condition that pure water or lubricant was applied to the colonic mucosa without washing and removal (i.e., the condition in this study). Under the condition in which pure water was applied, the friction coefficient increased by removing mucus. This indicates that mucus derived from the colon also plays a role as a lubricant to some extent. In contrast, under the lubricant-applied condition, no significant difference was observed in the friction coefficient depending on the presence or absence of mucus. Therefore, the interaction between mucus and lubricant may not need much consideration. Of course, further research is needed.

2. The paper uses water as a control group. However, considering that water might dilute the natural mucus of the colon, it is recommended to use a liquid closer to the natural state of the colon (such as saline or simulated colonic fluid) as a control group, or explicitly state this limitation and improve it in subsequent research.

Response: We appreciate your helpful comments. As the reviewer commented, saline may be preferable from a physiological perspective. However, during actual colonoscopy, tepid water rather than saline is injected into the colon to clean it and secure the visual field. Because it is assumed that our developed lubricants will be used in place of the tepid water in actual colonoscopy, pure water containing no salts was used as the solvent for the lubricants in this study. In addition, as a preliminary analysis, we measured DFC using saline instead of pure water and confirmed that the DFC values measured using pure water and saline were almost the same. The use of pure water as the control condition and lubricant solvent has been added to the discussion section (study limitations).

3. The paper utilized a colonoscope of a specific size and model to measure the DFC. However, in practice, colonoscopes of different models and sizes might exert varying pressures on the colon, leading to differences in the friction coefficient. It is recommended that the authors conduct the DFC tests across a range of colonoscope types and sizes to evaluate the influence of instrument size and design on lubrication. This would lend greater universality to the results.

Response: We appreciate your helpful and insightful suggestion. As you pointed out, the force applied to the colon mucosal surface may differ depending on the model and size of the endoscopes, and the friction coefficient varies with changes in load/force and speed. In this study, the friction coefficient was measured using our evaluation model shown in Figure S2 (new Figure S1), and the friction coefficients measured with a load of 100 g and a speed of 1.0 cm/s are presented in the manuscript. However, a prior analysis was conducted to measure the friction coefficient under different speed and load conditions. In this preliminary analysis, although the friction coefficient itself did change as the conditions were changed, the relationship between lubricant viscosity and the friction coefficient remained almost unchanged. Therefore, the optimal viscosity range for the lubricant remained unchanged.

In addition, preliminary friction evaluation was conducted using an endoscope with a smaller diameter than the colonoscope used in this study, and the friction coefficient was almost the same. However, it is desirable to conduct comprehensive evaluation using multiple endoscopes, and we are currently evaluating the friction coefficient between various endoscopes and the intestinal mucosa.

Reviewer #1 (Remarks to the Author):

All comments have been addressed, and I have no further comments on the manuscript.

Reviewer #2 (Remarks to the Author):

In the revised manuscript , the authors have addressed all my concern. I recommend to accept this paper in Communications Engineering

Responses to comments from the reviewers

=====

Reviewer #1:

All comments have been addressed, and I have no further comments on the manuscript.

Reviewer #2:

In the revised manuscript, the authors have addressed all my concern. I recommend to accept this paper in Communications Engineering

Response:

We would like to express our deep gratitude to all the reviewers for their positive evaluations.